# Establishment of a Novel Method for Spinal Discectomy Surgery in Elderly Rats in an In Vivo Spinal Fusion Model

**DOI:** 10.3390/mps4040079

**Published:** 2021-11-02

**Authors:** Katharina A. C. Oswald, Sebastian F. Bigdon, Andreas S. Croft, Paola Bermudez-Lekerika, Alessandra Bergadano, Benjamin Gantenbein, Christoph E. Albers

**Affiliations:** 1Department of Orthopaedic Surgery and Traumatology, Inselspital, Bern University Hospital, University of Bern, CH-3010 Bern, Switzerland; katharina.oswald@insel.ch (K.A.C.O.); sebastian.bigdon@insel.ch (S.F.B.); christoph.albers@insel.ch (C.E.A.); 2Tissue Engineering for Orthopaedics and Mechanobiology, Bone & Joint Program, Department for BioMedical Research (DBMR), Medical Faculty, University of Bern, CH-3008 Bern, Switzerland; andreas.croft@dbmr.unibe.ch (A.S.C.); paola.bermudez@dbmr.unibe.ch (P.B.-L.); 3Experimental Animal Center, Department for BioMedical Research (DBMR), Medical Faculty, University of Bern, CH-3008 Bern, Switzerland; alessandra.bergadano@dbmr.unibe.ch

**Keywords:** spine, spinal fusion, rat, animal experiment, rat anaesthesia, rat surgery, standard operating procedure, SOP, bone morphogenetic proteins

## Abstract

The rat model is a common model for intervertebral disc (IVD) and spinal research. However, complications remain challenging. Standard Operating Procedures (SOPs) are validated methods to minimize complications and improve safety and quality of studies. However, a SOP for rat spinal fusion surgery has been missing until now. Therefore, the aim of the study was to develop a SOP for spinal tail disc surgery in elderly Wistar rats (419.04 ± 54.84 g). An initial preoperative, intraoperative, and postoperative surgical setup, including specific anaesthesia and pain management protocols, was developed. Anaesthesia was induced by subcutaneous injection of a pre-mixture of fentanyl, midazolam, and medetomidin with the addition of 0.5% isoflurane in oxygen and caudal epidural analgesia. The surgery itself consisted of the fixation of a customized external ring fixator with ⌀ 0.8 mm Kirschner wires at the proximal rat tail and a discectomy and replacement with bone morphogenetic protein coated beta-tricalcium-phosphate carrier. The postoperative setup included heating, analgesia with buprenorphine, and meloxicam, as well as special supplementary food. Anaesthesia, surgery, and pain management were sufficient. In the presented optimized SOP, no animals developed any complications. A SOP for spinal surgery in elderly rats in an in vivo spinal fusion model was developed successfully. This novel protocol can improve transparency, reproducibility, and external validity in experimental rat spinal surgery experiments.

## 1. Introduction

Spinal fusion operations belong to the most common surgical interventions of the spine. In the United States, approximately 400,000 spinal fusions are performed annually with estimated associated costs at USD 32 billion per year [1,2]. Spinal fusion surgery is a common therapy option for various pathologic conditions of the spine, including degenerative disorders of the intervertebral discs, instabilities, trauma, tumors, infections, and deformities. Although fusion is an effective clinical treatment, the incidence of nonunion and/or pseudoarthrosis ranges between 5% and 35% in the lumbar spine [3,4]. Nonunion and pseudoarthrosis are severe complications as they can lead to persistent pain, instability, implant failure, and extended revision surgeries. Martin et al. reported that up to 23.6% of revision surgeries after spinal fusion are performed due to pseudoarthrosis [4,5]. The most common osteobiological additive for enhancement of spinal fusion is the bone morphogenetic protein 2 (BMP-2) [6]. It is currently used ‘off-label’ for different orthopaedic surgeries including spinal fusion, fractures, and long bone nonunion with a reported fusion rate of up to 99.1% when combined with autograft and bone graft extender [7,8,9]. However, the applied high doses of BMPs have often led to significant adverse events and pseudoarthrosis is reported to occur in up to 27% despite BMP-2 application [4,10]. Therefore, further investigation of osteological additives or drugs to enhance spinal fusion is highly medical, social, and socioeconomic. Current research focuses on investigating the effect of BMP-2 analogue L51P, which is reported to improve bone formation in combination with low doses of BMP-2 in vitro murine osteoblasts and in vivo femur mouse models [11,12,13,14]. The rat model is the most common in vivo animal model [15,16]. Additionally, Beckstein et al. reported that the mechanical performance of rat and human discs is very similar after normalization by disc height and area, which indicates largely conserved disc tissue properties across species [17]. The rat tail is considered to be a suitable proof-of-concept model to evaluate newer tissue-engineering strategies and to investigate the effect of agents to the integrity of spinal fusion [18,19,20]. However, complications such as wound dehiscence, wound infection, tail necrosis, insufficient anaesthesia, postoperative pain and weight loss, as well as sudden postoperative death, remain challenges in rat spine surgery [21,22,23]. Most spinal fusion studies are conducted on younger rats, aged 2–3 months, which corresponds to a 4 to 7-year-old human [24,25,26,27,28]. However, as most spinal fusion procedures are conducted in elderly patients, studies in elderly rats are warranted. Standard Operating Procedures (SOPs) are a validated method to minimize complications and improve safety in basic science projects, including animal experimentation. Additionally, they enhance the quality of the study in terms of transparency and reproducibility [29]. Although Martin et al. also investigated spinal fusion in elderly rats, a SOP for spinal surgery in these rats is missing up to this date [16]. 

This study aimed to develop a standard operating procedure for safe, transparent and reproducible preoperative, intraoperative, and postoperative surgical setup and procedures for spinal surgery in for in vivo rat spinal fusion models.

## 2. Experimental Design

We have developed a SOP for spinal surgery in elderly rats for spinal fusion models using bone morphogenetic protein (BMP) coated β-tri-calcium-phosphate (β-TCP) carrier and customized external ring fixator to allow compression. The PEEK fixator was developed by Martin et al. (2014) previously but for an artificial disc replacement strategy [15,16]. For this pilot, an initial preoperative, intraoperative, and postoperative surgical setup was developed for a pilot of 16 rats. After completion of the first pilot, a critical review of the setup was conducted. This included the preoperative, intraoperative, and postoperative setup and procedures. An optimization of the previous SOP was set up and used in a second pilot of 10 rats. This further optimized and approved SOP by the authorities is presented here. In our assay, we describe step by step the final spinal fusion model as summarized in Figure 1. This protocol will be very important to guide researchers who aim to investigate the enhancement of spinal fusion by drugs or growth factors in a rat model. The surgical and coating procedure is carried out under sterile conditions. All instruments are sterilized by autoclaving before the procedure. The study was conducted according to the guidelines of the Declaration of Helsinki and approved by the authorities and animal ethics committee of the canton of Bern (animal permit BE32/19). The study design follows the ARRIVE guidelines (https://arriveguidelines.org/, assessed 23 August 2021). 

### 2.1. Materials

Fentanyl 0.05 mg/mL (Sintetica S.A., Mendrisio, Switzerland).Midazolam 5 mg/mL (Sintetica S.A., Mendrisio, Switzerland).Medetomidine 1 mg/mL (Orion Corporation, Espoo, Finland).Meloxicam 5 mg/mL, injection solution (Boehringer Ingelheim GmbH, Basel, Switzerland).Meloxicam 0.5 mg/mL, oral suspension (Boehringer Ingelheim GmbH, Basel, Switzerland).Buprenorphin 0.3 mg/mL (Streuli Pharma SA, Uznach, Switzerland).Flumazenil 0.1 mg/mL (CPS Cito Pharma Services GmbH, Uster, Switzerland).Naloxon 0.4 mg/mL (OrPha Swiss GmbH, Küsnacht, Switzerland).Atipamezol 5 mg/mL (Orion Corporation, Espoo, Finland).Ropivacain 7.5 mg/mL (Fresenius Kabi AG, Kriens, Switzerland).Sodiumchloride 0.9% (Bichsel AG, Interlaken, Switzerland).Glucose 5% (B. Braun Medical AG, Sempach, Switzerland).Pentobarbital 300 mg/mL (Streuli Pharma SA, Uznach, Switzerland).Isofluran 99.9% (Provet AG, Lyssach, Switzerland).O_2_ 100%Sterile ophthalmic lubricant (Bausch & Lomb Swiss AG, Zug, Switzerland).Phosphate Buffered Solution (PBS).Bovine Serum Albumin (BSA) (Sigma–Aldrich Chemie GmbH, Buchs, Switzerland).OpSite transparent spray dressing (Smith & Nephew, Solothurn, Switzerland).Tissue culture-treated plate, 60 mm dish (TPP Techno Plastic Products AG, Schaffhausen, Switzerland).TOPIC spray (Vetoquinol AG, Bern, Switzerland).β-TCP carrier, ∅ 0.5 mm, h = 1.5 mm, 75% porosity (produced by Prof Marc Bohner, Robert Mathys Foundation, Bettlach, Switzerland) (Figure 2a).Growth Factors, BMP-2 5 mg/mL and L51P 5 mg/mL. Growth factors were delivered lyophilized and sterile, and were kindly provided (produced by Prof Walter Sebald, Department of Physiological Chemistry, University of Würzburg, Würzburg, Germany).Customized external ring fixator (Urs Rohrer, SITEM, Inselspital, University Hospital Bern, Bern, Switzerland) (Figure 2b).Customized hex wrench (Urs Rohrer, ARTORG Center, Inselspital, University Hospital Bern, Bern, Switzerland) (Figure 2b).Surgical instruments: scalpel holder, needle holder, anatomical forceps, surgical forceps, suture scissors, curette, sharp pliersat, Kerrison rongeur (B. Braun Medical AG, Sempach, Switzerland).Kirschner wires 0.8 mm (Synthes GmbH, Oberdorf, Switzerland).Carbon steel surgical blades 15 mm (Swann–Morton, Sheffield, United Kingdom).Ethanol 70%, 80%, 96%, 100%.Sterilium disinfection (Paul Hartmann AG, Heidenheim, Germany).Octenisept spray disinfection (Schülke & Mayr GmbH, Norderstedt, Switzerland).Sterile surgical gloves (Mölnycke Health Care AB, Göteborg, Sweden).Leukosilk silk plasters 2.5 cm × 5 m (BSN medical GmbH, Hamburg, Germany).Ethilon Sutures 4.0 (Johnson & Johnson Medical Ltd., Livingston, United Kingdom).Vicryl Sutures 5.0 (Johnson & Johnson Medical Ltd., Livingston, United Kingdom).Sterile compresses 5 × 5 cm, 10 × 20 cm (IVF Hartmann AG, Neuhausen, Switzerland).Sterile covers 20 × 20 cm (IVF Hartmann AG, Neuhausen, Switzerland).Cling FilmSyringes 1 mL, 2.5 mL, 5 mL, 10 mL (Terumo Corporation, Tokyo, Japan).25 G 16 mm, 22 G 30 mm, 20 G 30 mm needles (B. Braun Melsungen AG, Melsungen, Germany).Falcon Tubes, 50 mL (BD Falcon, New York, NY, USA).

### 2.2. Equipment

Biolaboratory safety class IISpecific-pathogen-free (SPF) animal facility with rodent operation roomAnaesthesia station (Vet Tech Solutions, Congleton, Great Britain).Rodent Surgical Monitor including heating pad, thermometer, and pulse oximeter, respiratory rate and heart rate (UNO B.V., PC Zevenaar, Netherlands).Bairhugger, heated air inflatable blanket (3 M Schweiz GmbH, Rüschlikon, Switzerland).Stryker System 8 cordless driver (Stryker, Selzach, Switzerland).

### 2.3. Animals

WISTAR CRL:WI (Han) Rats, retired breeders, 8–10 months old (Charles River Laboratories, Sulzfeld, Germany). A 10-month-old rat corresponds to an approximately 21.7-year-old human [27,28]. Martin et al. used 7–9-month-old rats for a rat tail spinal fusion model [16].

## 3. Procedure

### 3.1. Coating of β-TCP-Carrier

First, a stock solution with sterile H_2_O was prepared to reach a 1 mg/mL concentration for both growth factors, BMP-2 and L51P. Then, 4 mL PBS and 0.1% BSA was added to 1 mL of the stock solution to prepare a 0.2 mg/mL working solution. This solution was then further diluted with 1× PBS to reach the aimed concentration of growth factors per 50 µL solution according to the experimental groups.


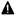
 **CRITICAL STEP** Do not vortex; shake by hand for approximately 10 min.

Apply 25 µL on one side of the selected growth factor in the desired concentration to the β-TCP carrier on a sterile Petri dish under sterile conditions. Let the growth factors diffuse in and let it dry for 1 h. Afterwards, turn the β-TCP carrier to the other side, repeat the application of 25 µL and let it dry again for 1 h. Store the carrier at 4 °C in closed, sterile Petri dishes for less than 48 h until use. 

### 3.2. Anaesthesia

Prepare the anaesthetic, analgesic, and antagonizing mixtures. The mixtures can be stored for up to 5 days at 4 °C until use. For ten rats, mix in a 50 mL Falcon tube each:

Anaesthetic Mixture: 1 mL Fentanyl (0.05 mg/mL) + 4 mL Midazolam (5 mg/mL) + 1.5 mL Medetomidine (1 mg/mL) + 3.5 mL NaCl (0.9%)

Subcutaneous application is 0.1 mL/100 g body weight (BW) of this mixture for a dose application of 0.005 mg/kg BW Fentanyl, 2 mg/kg BW midazolam and 0.15 mg/kg BW Medetomidine.

Analgesic Mixture: 2 mL Meloxicam (5 mg/mL) + 8 mL NaCl (0.9%) 

Subcutaneous application is 0.1 mL/100 g body weight (BW) of this mixture for a dose application of 1 mg/kg BW Meloxicam.

Anaesthesia Antagonizing Mixture (for surgical interventions): 1.66 mL Buprenorphin (0.3 mg/mL) + 20 mL Flumazenil (0.1 mg/mL) + 1.5 mL Atipamezol (5 mg/mL)

Subcutaneous application is 0.23 mL/100 g body weight (BW) of this mixture for a dose application of 0.05 mg/kg BW Buprenorphin, 0.2 mg/kg BW Flumazenil and 0.75 mg/kg BW Atipamezol.

For surgical procedures, Fentanyl is displaced postoperatively by buprenorphine to ensure full analgesia. If anaesthesia is used for X-ray read-outs, Buprenorphine should be replaced with Naloxone and the following mixture used:

Anaesthesia Antagonizing Mixture (for non-surgical interventions): 3 mL Naloxon (0.4 mg/mL) + 20 mL Flumazenil (0.1 mg/mL) + 1.5 mL Atipamezol (5 mg/mL) + 5.5 mL NaCl (0.9%)

Subcutaneous application is 0.3 mL/100 g body weight (BW) of this mixture for a dose application of 0.12 mg/kg BW Naloxon, 0.2 mg/kg BW Flumazenil and 0.75 mg/kg BW Atipamezol.


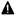
 **CRITICAL STEP** Anaesthetic mixture should be applied subcutaneously in loose restrained handling into the nuchal fold. Make sure to vent the needle prior to injection and use a different injection needle (22 G 30 mm) than the needle used for drawing up the anaesthetic mixture to avoid a blunt needle tip. Directly after injection, release loose restrained handling to avoid reflux of the fluid (Figure 3a,b).

After application of the anaesthetic, wait 5–10 min until full surgical anaesthesia is reached in the rat. Test surgical depth of the anaesthesia with the withdrawal reflex of forelimb pinching (with the fingers or forceps). These tests should be negative.

Apply sterile ophthalmic lubricant in the rat’s eyes to prevent eye damage from the surgical lamp and dry out.

Locoregional analgesia is performed by epidural injection of a local anaesthetic. First, clip the hair on the dorsal side of the tail and thereafter disinfect with Octenisept. Then, inject 0.12 mL Ropivacaine 0.75% with a 25 G 16 mm needle into the epidural space of Cauda 1/2. 

MMF anaesthesia, which lasts for approximately 45–60 min, length is supplemented with 0.5–1 Vol.-% Isoflurane in 0.5 l/min O_2_ (100%) applied over a customized rat face mask using a non-rebreathing anaesthesia circuit.

Body temperature is monitored by placing the thermal probe orally to avoid displacements due to surgical manipulations. Oxygen saturation and heart rate is supervised with a pulse oximeter for the paw, using the Rodent Surgical Monitor. Adjust Isoflurane/ Oxygen flow and temperature of the integrated heated platform according to vital parameters throughout the surgery.

### 3.3. Surgical Setup

The anaesthetized animals are positioned in sternal recumbency for the surgical setup on the heat platform in the centre of a clean, separate table. The tail should lay freely outside the platform. The anaesthetist and the anaesthesia machine should be positioned close to the rat’s head and the surgeons close to the rat’s tail, respectively. Two surgeons are recommended for an optimal outcome. Surgical instruments are placed next to the heat platform on a sterile cover (Figure 3c). The rat is instrumented with face mask, thermometer and pulsoximeter (Figure 3d). The animal is loosely wrapped twice with cling film around the platform to secure the position of the animal throughout the surgery and keep normothermia. The whole tail is disinfected with alcohol-based skin disinfection (Octenisept) and pulled through the sterile coverage in a sterile manner (Figure 3e). Then, the surgeons disinfect their hands with alcohol-based disinfection (Sterilium) and put on sterile gloves.

### 3.4. Surgical Procedure

First, disinfect the surgical site again with alcohol-based skin disinfection. Next, use a surgical pen to draw the length of the skin incision over spinal segment Cauda 4/5 (2 cm long). For optimal overview, mark the two adjacent intervertebral disc spaces with a 25 G 16 mm needle. Mount the external ring fixator by inserting k-wires percutaneously with the cordless driver. Two Kirschner (k)-wires will be inserted on the upper adjacent vertebra and the other two k-wires through the lower adjacent vertebra from left to right (Figure 1B and Figure 3f,g). Begin with the most cranial k-wire and mount the fixator around the first k-wire. Use the precut holes of the fixator as guidance to insert the remaining k-wires from cranially to caudally (Figure 3f,g). Further visualization of this protocol is given in the Appendix A.


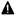
 **CRITICAL STEP** Insert the k-wires in a slightly posterolateral direction to avoid damaging the vessels running laterally to the vertebrae.

After mounting of the fixator, distract the segment by tightening the nut with the hex wrench. Perform an approximately 2 cm long midline skin incision over Cauda 4/5. Sharply dissect through the fascia. This allows the identification of the two posterior longitudinal tendons, which remain untouched. The surgery is continued with blunt midline dissection between posterior longitudinal tendons to midline muscle followed by sharp dissection of the muscle in the midline. Identify the disc and adjacent vertebra and perform a sharp submuscular dissection around the disc. Remove the disc sharply from the endplates of the adjacent vertebra. The Nucleus pulposus material is removed with a Kerrison rongeur, and the endplate cartilage is debrided with a small, sharp curette (Figure 1C and Figure 3h). Irrigate the disc niche with 10 mL NaCl. Afterwards, insert the previously coated β-TCP carrier cautiously into the disc niche to avoid midline breakage of the carrier (Figure 3i). Compress the disc niche using the external ring fixator threaded rods to press-fit the carrier between the two adjacent vertebrae. The rods were tightened with the customized hex wrench (see also Appendix A). Close the wound in a two-step procedure and apply sterile OpSite wound dressing (Figure 3j,k).


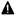
 **CRITICAL STEP** Wound closure should be performed in a two-step process with subcutaneous closure using 4.0 Vycril and cutaneous closure with 5.0 Ethilone to ensure complete wound closure and prevent animal wound biting and wound infection.

Cut the k-wires with the sharp pliers at the level of the ring fixator (Figure 3g,k). Make sure not to leave sharp ends.

### 3.5. Postoperative Regimen

After the surgery and completed wound closure, apply the anaesthetic antagonizing mixture and the analgesic mixture directly subcutaneously with a 22 G 30 mm needle. Animals should recover in a clean cage, where they will wake up 3–5 min after injection. These cages should be placed under a Bair hugger for at least 2 h postoperatively to achieve normothermia. Make sure to give the rats access to the analgesic water directly postoperatively. It is recommended to start analgesic water treatment 24 h preoperatively to give the animals time to adjust to the new water taste. Analgesic water is given for at least 48 h postoperatively and longer as needed.

Analgesic water mixture (per rat for 72 h):360 mL autoclaved H_2_O+ 10 mL Glucose (5%)+ 6 mL Buprenorphine (0.3 mg/mL)

Apply analgesic water over a regular drinking bottle with free access to the animal. Give additional analgesia with 1 mg/kg BW Meloxicam per os/day for 5 consecutive days. Animals should be monitored 5 times per day for 5 postoperative days. There, a health and overall wellbeing check, as well as a pain score sheet, should be performed and additional analgesia with Meloxicam per os provided as needed. The pain score sheet should include the following categories: body weight and food intake, demeanour, deambulation, facial expression, wound condition, the position of the external fixator. If there is an increased need for analgesia, monitoring time points should be increased accordingly. Critical care feed (Omnivore, Emeraid^®^) was provided in the bottom of the cage in the first 3 postoperative days as a source of calories, essential nutrients and for supporting hydration.


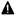
 **CRITICAL STEP** Check on the correct positioning of the external ring fixator and closed wounds during the animal checkups.

### 3.6. Euthanasia

After completion of the project, animals are euthanized using pentobarbital.

Euthanasia mixture: 1 mL Pentobarbital (300 mg/mL) + 9 mL NaCl (0.9%)

Intraperitoneal application is 0.05 mL/100 g BW of this mixture with a 25 G 16 mm needle for a dose application of 150 mg/kg BW Pentobarbital. Death is confirmed by clinical signs (respiratory and heart arrest) and exsanguination.

## 4. Expected Results

This SOP describes the pre-, peri-, and postoperative setup and procedures for spinal surgery in elderly rats for in vivo spinal fusion models.

### 4.1. Effectiveness of the Protocol

#### 4.1.1. Anaesthesia

Anaesthesia was adequate in both conducted pilots. Animals weighed 419.04 ± 54.84 (310–515) g (38.46% female) at surgery. In the first pilot, a concentration of 1 mg/mL midazolam was used for the anaesthetic mixture. As the concentration of 1 mg/mL resulted in a high amount of mixture (0.94 ± 0.13 (0.68–1.2) mL) to be applied subcutaneously, we changed the concentration of midazolam to 5 mg/mL. Then, 0.29 ± 0.05 (0.23–0.38) mL was applied. Animals fell asleep after 7.50 ± 2.72 (4–13) min. Rats were 55.30 ± 14.78 (44–84) min under surgical anaesthesia. The midazolam–Medetomidin–Fentanyl mix provided a reliable and reproducible anaesthesia of good quality. None of the animals woke up before the end of the surgery. The supplementation with 0.5–1% isoflurane was a hypnotic adjuvant and would allow for rapid deepening in the anaesthetic depth or prolongation of the anaesthesia beyond 1 h if required. This was not necessary in any of the animals. No animal died during surgery. Rats woke up after application of the antagonizing mixture (0.98 ± 0.14 (0.7–1.2) mL) after median of 6.00 [Interquartilrange 4.76–9.50; Minimum 4–Maximum 38] min.

#### 4.1.2. Surgical Procedure

Surgery was sufficient in both pilots. Mean operation time was 27.09 ± 6.28 (20–40) min. In the postoperative X-rays, 18 (69.23%) rats had optimal postoperative radiographic outcome concerning the position of k-wires and β-TCP carrier. Eight animals had sufficient, yet non-optimal radiographic outcome. In three animals, the β-TCP carrier was broken in the midline and positioned slightly too dorsally in another 3 animals. An intradiscal position of one of the four k-wires was observed in X-rays in two animals: However, in these two animals the fixator was sufficiently stable during the study period of 12 weeks. During the first pilot, disinfection was performed one time on the surgical site and all surgical covers were positioned underneath the animal. As two rats suffered from wound infection postoperatively, we changed the protocol to a three-step disinfection process as described above and placed the rat’s tail through the sterile cover to avoid wound contamination with the animal’s fur and decline contamination with animal skin bacteria. Afterwards, no animal developed wound infection in the second pilot. Initially, k-wires were inserted parallel to the horizontal line. As three animals developed tail necrosis, the insertion direction of the wires was changed to a posterolateral direction to avoid damaging the tail vessels, which run on the dorsal lateral tail site. Subsequently, no animal developed tail necrosis in the second pilot. As two animals suffered from wound dehiscence during the first pilot, where only a cutaneous wound closure was performed, a two-step wound closure with added subcutaneous sutures was chosen for the second pilot. There, no animal developed wound dehiscence. 

#### 4.1.3. Pain Management

Epidural anaesthesia was easy to perform. Motor and sensitive block were present at recovery and for at least 2 h postoperatively. 

Pain management was sufficient during both conducted pilots. None of the animals refused to drink the anaesthetic water. None of the animals reached an endpoint with need for subsequent euthanasia due to pain.

#### 4.1.4. Complications

During the first pilot of 16 animals, seven rats developed complications with subsequent euthanasia in the first five postoperative days. Two animals suffered from wound dehiscence, another two rats from wound infection. Three animals developed tail necrosis, one of those had critical weight loss and another one suffered from tail nerve injury. Additionally, another animal died directly postoperatively for unknown reasons. In the second pilot, in which the optimized and presented protocol was used, no animals were lost. None of the animals developed neither wound infection, nor tail necrosis, nor critical weight loss. 

#### 4.1.5. Euthanasia

Euthanasia was sufficient in both conducted pilots. Death was confirmed 11.33 ± 5.83 (5–27) min after application of 2.50 ± 0.50 (1.8–3.34) mL of the pentobarbital mixture.

### 4.2. Data Analysis and Interpretation

#### 4.2.1. X-rays and µCT

True lateral and oblique anterior–posterior X-rays are conducted at the day of surgery, three, six, and 12 weeks postoperatively at 25 kV and 10 s acquisition time (MX-20, Faxitron X-Ray Corporation, Edimex, Le Plessis, France) to obtain a reference image and double-check the correct placement of the carrier and the k-wires. Animals are under full anaesthesia at Day 0, 3, and 6 weeks postoperatively and will be sacrificed 12 weeks postoperatively. At 12 weeks, µCT (MicroCT40, SCANCO Medical AG, Brüttisellen, Switzerland; with the built-in software from SCANCO (SCANCO Module 64-bit; V5.15)) is conducted. Examples for X-ray and µCT data are shown in Figure 4. Analysis of spinal fusion on conventional X-rays will be done according to the established Bridwell criteria [30]. Due to the lack of a validated method to analyse spinal fusion qualitatively on (µ)CT data, and adapted method, based on the Bridwell criteria, can be used (Table 1).

Quantification of new bone formation on µCT images is performed using OsiriX DICOM Viewer (Pixmeo, Bernex, Switzerland). To distinguish between new bone formation and carrier material, tissue is segmented into 3 tissue types based on their greyscale, i.e., <200 Hounsfield unit (HU) for soft tissues, between 200 HU and 360 HU for low density mineralised tissues (LDMT) and >360 HU for high density mineralised tissues (HDMT). The analysis is performed at the highest resolution with a voxel size of 6 μm.

#### 4.2.2. PMMA Histology and Immunohistology

To quantify bone formation 12 weeks postoperatively, samples are embedded in Polymethylmethacrylat (PMMA). Thereupon, they are cut into 400 μm ground sections, polished to 200 µm ground sections and finally stained with MacNeal’s tetrachrome [31,32]. Microphotographs are taken using a Nikon Eclipse E800 microscopy system (Nikon Inc., Switzerland, Egg, Switzerland). Bone formation and implant turnover is determined by the ImageJ trainable segmentation plugin; Waikato Environment for Knowledge Analysis (WEKA) automated segmentation histomorphometry [33] on 4 serial McNeal tetrachrome-stained ground sections per implant. The sum of the surface area of bone and carrier will be determined for each animal (four sections/animal). The total bone surface area and implant surface area per treatment group will be computed. For immunohistochemical staining (Osteocalcin (OCC) (FL-110:sc-30044; Santa Cruz Biotechnology, Paso Robles, CA, USA), bone sialoprotein (BSP), and osteonectin (ONC)), Paraffin embedding will be conducted after decalcification over 6 months. The intensity of the immunohistochemical staining of will be classified for all and antibodies will be categorized as mild (+), moderate (++), or intense (+++).

### 4.3. Data Application to Human Clinical Trials

Khan et al. reported that the rat model is a suitable proof-of-concept model to evaluate newer tissue-engineering strategies and to investigate the effect of agents to the integrity of spinal fusion. It is also suitable for safety, feasibility and efficacy studies [18,19,34]. Furthermore, the growth-factor BMP-2 and tissue engineered, calcium-tri-phosphate-based, allografts are already used in clinical practice [7,8,9]. As L51P is a very similar, yet advantageous, growth factor, it seems that our model could be applied to humans and be investigated in clinical trials [35].

### 4.4. Comparison of Surgical Technique

This is the first study elaborating a standard operating procedure for spinal surgery on the rat’s tail. Martin et al. inserted the carrier in a lower segment (Cauda 8/9) than in the present study [16]. In addition, a disc degeneration model used more caudal segments such as C6/7 or C8/9 [26]. Ding et al. performed surgery on cranial vertebrae in an osteoporosis degeneration model [25]. We have chosen a proximal caudal segment—Cauda 4/5—due to its excellent accessibility and fast detection of the correct segment as well as the larger disc space. Furthermore, as nonunion is mostly a challenge in the lumbar spine in humans, cervical vertebrae would not have been applicable. Due to the testicles of male rats, operating on a more proximal segment could have led to impairment of defecation due to space narrowing through the ring fixator. Besides the presented study, only Martin et al. used an external ring fixator with k-wires and a TCP carrier for their model. In contrast to Martin et al., we have fixated the ring fixator with two—instead of one—k-wires cranially and caudally to the operated segment to provide also rotational stability [16]. The mounting of the fixator and the insertion of the k-wires were similar in both studies (Martin 2014). Concluding, the rat’s spine is a common model for disc degeneration and spinal fusion animal experimentation. However, this is the first standard operating procedure for spinal surgery on the rat’s tail in spinal fusion models.

## 5. Reagents Setup

Phosphate Buffered Solution (PBS): 2.0 g KCL + 2.4 g Kh_2_PO_4_ + 14.2 g Na_2_HPO_4_ − 2H_2_O + 80 g NaCl

Dissolve in 88 mL distilled water, adjust to pH 7.4 and fill with distilled water to 1000 mL. To reach normal PBS concentration, mix 100 mL of the solution with 900 mL distilled water.

## Figures and Tables

**Figure 1 mps-04-00079-f001:**
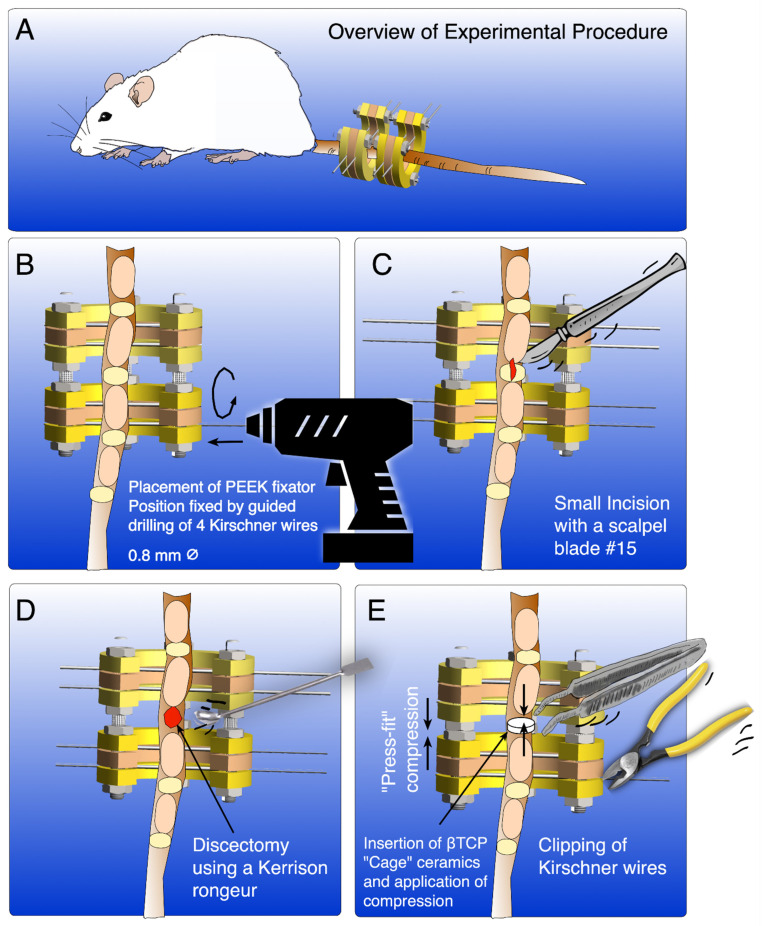
Representing scheme illustrating the SOP for spinal surgery in elderly rats for spinal fusion models. (**A**). overview of fixator mounted on adult Wistar rat (**B**) Mounting of fixator extern by drilling four Kirschner wires through two adjacent vertebrae (**C**). Midline incision using a small scalpel blade, e.g., #15 (**D**) discectomy with a Kerrison rongeur (**E**) insertion of βTCP carrier (± coated with BMPs). This procedure was followed by press-fit compression using the fixator extern, finalized by a two-step wound closure and clipping of the Kirschner wires.

**Figure 2 mps-04-00079-f002:**
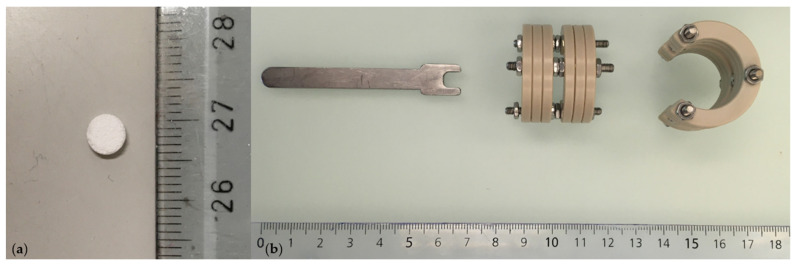
(**a**) TCP carrier, (**b**) Customized external ring fixator and customized hex wrench. The external fixator is based on the study by Martin et al. (2014) [16].

**Figure 3 mps-04-00079-f003:**
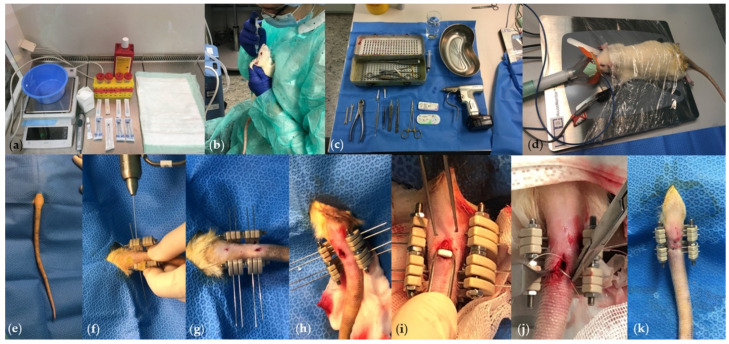
(**a**) Anaesthesia setup with medication mix, scale, disinfection, hair clipper, injection syringes, and needles. (**b**) Anaesthetic mixture injection in loose restrained handling into the nuchal fold. (**c**) Surgical instruments including k-wires, cordless driver, sharp pliers at, curette, forceps, Kerringson rongeur, scalpel, and needle holder. (**d**) Surgical setup with a clean table, heat platform, pulse oximeter, thermometer, face mask for isoflurane application and cling film to secure the position of the rat. (**e**) Sterile coverage. (**f**) Application of external ring fixator with four Kirschner wires. (**g**) Mounted external fixator. (**h**) Midline incision on the rat’s tail at Cauda 4–5 and discectomy. (**i**) Insertion of coated β-TCP carrier. (**j**) Two-step wound closure. (**k**) Wire shortening to ring fixator level.

**Figure 4 mps-04-00079-f004:**
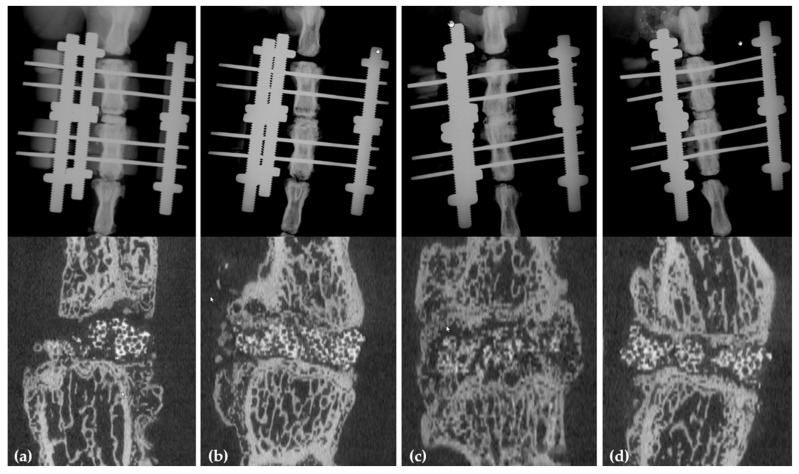
Imaging of exemplary X-rays and µCT 12 weeks postoperatively with (**a**) (Group 1: βTCP + PBS, negative control) showing no fusion and incomplete reabsorption of the carrier; (**b**) (Group 2: βTCP + 1 µg BMP-2) showing partial fusion with integration of the carrier and partial bridging callus formation; (**c**) (Group βTCP + 10 µg BMP-2) showing complete fusion with bridging callus, integration and incomplete grow-through of the carrier with bone; (**d**) (Group βTCP + 10 µg L51P) showing no fusion and sparsely reabsorption of the carrier.

**Table 1 mps-04-00079-t001:** Parameters for radiographic µCT evaluation.

Parameter			
Fusion	none	incomplete	complete
On-growth on proximal endplate in % of endplate surface	none	<50	>50
On-growth on distal endplate in % of endplate surface	none	<50	>50
In-growth through carrier in % of carrier surface	none	<50	>50
Bridging callus formation in % of endplate surface	none	<50	>50
Carrier absorption in % of carrier surface	none	<50	>50

## Data Availability

The data can be obtained from the corresponding author.

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
