# Peer review of "Establishment of a Novel Method for Spinal Discectomy Surgery in Elderly Rats in an In Vivo Spinal Fusion Model"

_mps, 2021, doi:10.3390/mps4040079_

Round 1

Reviewer 1 Report

I congratulate the authors for the well-written paper.

Experimental animal models are a good way to train surgery and to test new materials, even though I this specific case the biomechanics of the intervertebral lumbar discs cannot be compared between rats and humans.

The protocol is clear and helps most of all to guarantee the respect for animal welfare.

Author Response

Reviewer 1.

Q1: I congratulate the authors for the well-written paper.

Experimental animal models are a good way to train surgery and test new materials, even though the biomechanics of the intervertebral lumbar discs cannot be compared between rats and humans in this specific case.

The protocol is straightforward and helps, most of all, to guarantee respect for animal welfare.

R: Thank you very much for your positive review of our manuscript.

It is correct, that the biomechanics of the rat’s IVD is not 100 percent congruent to the human’s IVD. However, Beckstein et al. reported that the mechanical performance of rat and human discs is very similar after normalization by disc height and area, which indicates largely conserved disc tissue properties across species (Beckstein et al Spine2008).

Also, the rat tail is reported to be a suitable proof-of-concept model to evaluate newer tissue-engineering strategies and to investigate the effect of agents to the integrity of spinal fusion (Khan et al. Biomaterials 2004; Boden et al. Spine 1998; Salamon et al. J Spinal Disord Tech 2003).

Additionally, an initial animal model inherits advantages compared to clinical trials such as minimal variation of the study population, surgical technique, follow-up and local environment. Furthermore, outcome variable analysis can be precisely conducted through e.g. histology, which is not possible in clinical studies (Schimandle et al. Spine 1994, Khan et al. Biomaterials 2004).

We have amended the manuscript accordingly (p.2, l. 58-62).

Reviewer 2 Report

Comments to the Author

I thank you for the opportunity to review your paper.

This paper describes a new methodology in the spinal fusion model of aged rats.

The authors developed a Standard Operating Procedures (SOP) for spinal tail disc surgery in elderly Wistar rats (419.04 ± 54.84 g). The surgery itself consisted of the fixation of a customized external ring fixator with ⌀0.8 mm Kirschner wires at the proximal rat tail and a discectomy and replacement with bone morphogenetic protein-coated beta-tricalcium-phosphate carrier. This novel protocol can improve transparency, reproducibility, and external validity in experimental rat spinal surgery experiments.

This is a rat study, and there remains a big question of whether it can be applied to humans.

My specific comment is below.

-“WISTAR CRL:WI (Han) Rats, old breeders, 8 -10 months old (Charles River Laboratories, Sulzfeld, Germany). “How old is this rat in humans?

-Line 63, “However, an SOP for spinal surgery in elderly rats is missing up to this date. “Please describe comparative data between conventional rats and elderly rats or previous references.

-The description of the method and results confuses me. The method and results should be easy for the reader to understand.

-Please be more specific about the surgical procedure than about anesthesia.

-“18 (69.23 %) rats had optimal postoperative radiographic outcome concerning the position of k-wires and β-TCP carrier. In three animals, the β- TCP carrier was broken and malpositioned in another 3 animals. “The failure of 30% contradicts the conclusion.

Author Response

Reviewer 2.

Q1: I thank you for the opportunity to review your paper.

This paper describes a new methodology in the spinal fusion model of aged rats.

The authors developed a Standard Operating Procedures (SOP) for spinal tail disc surgery in elderly Wistar rats (419.04 ± 54.84 g). The surgery itself consisted of the fixation of a customized external ring fixator with 0.8 mm Kirschner wires at the proximal rat tail and a discectomy and replacement with bone morphogenetic protein-coated beta-tricalcium-phosphate carrier. This novel protocol can improve transparency, reproducibility, and external validity in experimental rat spinal surgery experiments.

This is a rat study, and there remains a big question of whether it can be applied to humans.

R: Thank you very much for your excellent suggestions and comments.

As you point out, it is of high relevance whether our rat tail spinal fusion model can be applied to humans. It is correct, that a rat model is not 1:1 applicable to humans, as these two “species” are of different size, different anatomy and they have a different way of walking which leads to different biomechanics.   

However, Khan et al. reported that the rat model is a suitable proof-of-concept model to evaluate newer tissue-engineering strategies and to investigate the effect of agents to the integrity of spinal fusion (Khan et al Biomaterials2004; Boden et al. Spine 1998; Salamon et al. J Spinal Disord Tech 2003). It is also suitable for safety, feasibility and efficacy studies.

Additionally, Beckstein et al. reported that the mechanical performance of rat and human discs is very similar after normalization by disc height and area, which indicates largely conserved disc tissue properties across species (Beckstein et al Spine 2008)

Furthermore, the growth-factor BMP-2 and tissue engineered, calcium-tri-phosphate based, allografts are already used in clinical practice (Carragee et al The Spine Journal 2011; Parajon et al. Neurosurgery 2017; Simmonds et al. Annals of internal medicine 2013). As L51P is a very similar, yet advantageous, growth factor, which can be produced industrially, it seems like our model can be applied to humans.

We have amended this in the manuscript accordingly (p.12, l. 451-458).

Q2: WISTAR CRL:WI (Han) Rats, old breeders, 8 -10 months old (Charles River Laboratories, Sulzfeld, Germany). “How old is this rat in humans?

R: Thank you for pointing out this important aspect.

The median natural lifespan for the WISTAR rat is 850 days for males and 900 days for females (Ghirardi et al. Exp Gerontol 1995). There are several studies investigating the correlation between the rat and human age (Peckham The laboratory rat 1979; Gittes et al. Urol Clin North Am 1986; Iandoli et al. Acta Cir Bras 2000; Klee et al. J Urol 1990). However, it remains challenging to determine an exact age, as rats have an accelerated childhood in comparison to humans. They develop rapidly during infancy and become mature at about 6 weeks of age, whereas humans typically do not enter puberty until they are eleven years of age or older. Most studies state that an average of 13.2 rat days correspond to one human day (Quinn et al. Nutrition 2005; Sengupta et al. Int J Prev Med 2013). Therefore, a 10-month year old WISTAR rat would be a 21.7-year-old human ((10x30)/13.2 = 21.7)) and at end of the study, rats would be a 29.5 year old human ((13x30)/13.2 = 29.5)). Martin et al. used 7 -9 month old rats in a spinal fusion model (Martin et al. Acta Biomat 2014). We used these retired breeders to endorse the 3Rs.

We have amended the manuscript accordingly (p.2 l. 65-68, p.5. l. 172-175).

Q3: Line 63, “However, an SOP for spinal surgery in elderly rats is missing up to this date. “Please describe comparative data between conventional rats and elderly rats or previous references.

R: Thank you for pointing this out. It is crucial that references for scientific statements are provided. We have amended it accordingly in the manuscript (p.2. l. 65-73).

Q4: The description of the method and results confuses me. The method and results should be easy for the reader to understand.

R: Thank you for pointing this out. It is of high importance that the methods and results section is easy for the reader to understand. We have clarified the wording in the method and results section (p. 5 - 12).

Q5: Please be more specific about the surgical procedure than about anesthesia.

R: Thank you for pointing this out. Specific details for surgical procedures are very important. We have amended it accordingly in the manuscript and provided more details for the surgical procedure (p.7, 8. l. 263 - 304). We believe that the anesthetic procedure is a very important part of the SOP as a sufficient anesthesia is crucial for conducting surgery and for the animal’s wellbeing. Additionally, it seems like surgical animals studies are often conducted by researchers with a surgical training background, who might have less experience with anesthesia than with surgeries, also if trained correctly by laboratory animal courses. Therefore, details about the anesthesia seems to be crucial information for safe and efficient animal surgery.

Q6: 18 (69.23 %) rats had optimal postoperative radiographic outcome concerning the position of k-wires and β-TCP carrier. In three animals, the β- TCP carrier was broken and malpositioned in another 3 animals. “The failure of 30% contradicts the conclusion.

R: Thank you for pointing out this important aspect. If a high failure rate were present, a positive conclusion could not be drawn. However, with non-optimal radiographic outcome, we did not refer to failure of the procedure. Concerning k-wires, it reports optimal positioning of 3 of 4 k-wires and of 1 k-wire in the disc space. We consider this as sufficient, as stability is guaranteed, yet not optimal positioning. Additionally, as the TCP carrier is used as a coated carrier with no primary main segment stability function, and the growth factors can still migrate to the bone tissue in presence of a broken carrier or a slightly too dorsal position, we consider this as non-optimal yet sufficient radiographic outcome and therefore not as failure of the procedure.

We have amended the manuscript accordingly (p.9. l. 366-370).

Reviewer 3 Report

Authors introduce a standardized operative protocoll /SOP for spinal surgery in elderly rats in an in vivo spinal fusion model. Pre, intra and postoperative setup were standardized. The surgery itself consisted of the fixation of a customized external ring fixator with ⌀ 0.8 mm 25
Kirschner wires at the proximal rat tail and a discectomy and replacement with bone morphogenetic 
protein coated beta-tricalcium-phosphate carrier. 

Controlled trials on rats for investigation of effects of bone-morphogenetic proteins are important as a first basic research step to introduction of these proteins to clinical practice to promote fusion. That is why it is important to have an SOP when performing surgery, and it is astonishing that SOPs previously did not exist. The manuscript is equiped with very nicely done figures and procedure photos. For general readership, it would be of use to discuss several studies where rats were used in this same way, and to try to make a comparison of their surgical methods described to the SOP of the authors; for example:

Wang L, Cui W, Kalala JP, Hoof TV, Liu BG. To investigate the effect of osteoporosis and intervertebral disc degeneration on the endplate cartilage injury in rats. Asian Pac J Trop Med. 2014 Oct;7(10):796-800. doi: 10.1016/S1995-7645(14)60139-5. PMID: 25129463.

Liao JC. Cell Therapy Using Bone Marrow-Derived Stem Cell Overexpressing BMP-7 for Degenerative Discs in a Rat Tail Disc Model. Int J Mol Sci. 2016 Jan 22;17(2):147. doi: 10.3390/ijms17020147. PMID: 26805824; PMCID: PMC4783881.

If possible, provide a video of the procedure for better understanding.

Author Response

Reviewer 3.

Q1: Authors introduce a standardized operative protocoll /SOP for spinal surgery in elderly rats in an in vivo spinal fusion model. Pre, intra and postoperative setup were standardized. The surgery itself consisted of the fixation of a customized external ring fixator with 0.8 mm 25

Kirschner wires at the proximal rat tail and a discectomy and replacement with bone morphogenetic

protein coated beta-tricalcium-phosphate carrier.

Controlled trials on rats for investigation of effects of bone-morphogenetic proteins are important as a first basic research step to introduction of these proteins to clinical practice to promote fusion. That is why it is important to have an SOP when performing surgery, and it is astonishing that SOPs previously did not exist. The manuscript is equiped with very nicely done figures and procedure photos. For general readership, it would be of use to discuss several studies where rats were used in this same way, and to try to make a comparison of their surgical methods described to the SOP of the authors; for example:

Wang L, Cui W, Kalala JP, Hoof TV, Liu BG. To investigate the effect of osteoporosis and intervertebral disc degeneration on the endplate cartilage injury in rats. Asian Pac J Trop Med. 2014 Oct;7(10):796-800. doi: 10.1016/S1995-7645(14)60139-5. PMID: 25129463.

Liao JC. Cell Therapy Using Bone Marrow-Derived Stem Cell Overexpressing BMP-7 for Degenerative Discs in a Rat Tail Disc Model. Int J Mol Sci. 2016 Jan 22;17(2):147. doi: 10.3390/ijms17020147. PMID: 26805824; PMCID: PMC4783881.

R: Thank you very much for your positive revision of our manuscript.

We agree that comparing our surgical method with that of others is of use and have amended this accordingly in the manuscript (p. 12 l. 459 - 476). As the manuscript of Wang et al. (2014) has been retracted by the journal due to plagiarism issues, we did not include it in the discussion.

Q2:  If possible, provide a video of the procedure for better understanding.

R: Thank you very much for bringing up the high-value point of providing an educational video of the whole procedure for better understanding. We have followed your recommendation and have produced a video of the surgical procedure (additional files).